# Fast Distributed $k$-Center Clustering with Outliers on Massive Data

**Gustavo Malkomes,   Matt J. Kusner,   Wenlin Chen**
Department of Computer Science and Engineering
Washington University in St. Louis
St. Louis, MO 63130
{luizgustavo,mkusner,wenlinchen}@wustl.edu

**Kilian Q. Weinberger**
Department of Computer Science
Cornell University
Ithaca, NY 14850
kqw4@cornell.edu

**Benjamin Moseley**
Department of Computer Science and Engineering
Washington University in St. Louis
St. Louis, MO 63130
bmoseley@wustl.edu

## Abstract

Clustering large data is a fundamental problem with a vast number of applications. Due to the increasing size of data, practitioners interested in clustering have turned to distributed computation methods. In this work, we consider the widely used $k$-center clustering problem and its variant used to handle noisy data, $k$-center with outliers. In the noise-free setting we demonstrate how a previously-proposed distributed method is actually an $O(1)$-approximation algorithm, which accurately explains its strong empirical performance. Additionally, in the noisy setting, we develop a novel distributed algorithm that is also an $O(1)$-approximation. These algorithms are highly parallel and lend themselves to virtually any distributed computing framework. We compare each empirically against the best known sequential clustering methods and show that both distributed algorithms are consistently close to their sequential versions. The algorithms are all one can hope for in distributed settings: they are fast, memory efficient and they match their sequential counterparts.

## 1   Introduction

Clustering is a fundamental machine learning problem with widespread applications. Example applications include grouping documents or webpages by their similarity for search engines [30] or grouping web users by their demographics for targeted advertising [2]. In a clustering problem one is given as input a set $U$ of $n$ data points, characterized by a set of features, and is asked to cluster (partition) points so that points in a cluster are similar by some measure. Clustering is a well understood task on modestly sized data sets; however, today practitioners seek to cluster datasets of massive size. Once data becomes too voluminous, sequential algorithms become ineffective due to their running time and insufficient memory to store the data. Practitioners have turned to distributed methods, in particular MapReduce [13], to efficiently process massive data sets.

One of the most fundamental clustering problems is the $k$-center problem. Here, it is assumed that for any two input points a pair-wise distance can be computed that reflects their dissimilarity (typically these arise from a metric space). The objective is to choose a subset of $k$ points (called *centers*) that give rise to a clustering of the input set into $k$ clusters. Each input point is assigned to the cluster defined by its closest center (out of the $k$ center points). The $k$-center objective selects these centers to minimize the farthest distance of any point to its cluster center.

The $k$-center problem has been studied for over three decades and is a fundamental task used for exemplar based clustering [22]. It is known to be NP-Hard and, further, no algorithm can achieve a $(2-\epsilon)$-approximation for any $\epsilon > 0$ unless P=NP [16, 20]. In the sequential setting, there are algorithms which match this bound achieving a 2-approximation [16, 20].

The $k$-center problem is popular for clustering datasets which are not subject to noise since the objective is sensitive to error in the data because the worst case (maximum) distance of a point to the centers is used for the objective. In the case where data can be noisy [1, 18, 19], previous work has considered the $k$-centers with outliers problem [10]. In this problem, the objective is the same, but additionally one may discard a set of $z$ points from the input. These $z$ points are the outliers and are ignored in the objective. Here, the best known algorithm is a 3-approximation [10].

Once datasets become large, the known algorithms for these two problems become ineffective. Due to this, previous work on clustering has resorted to alternative algorithmics. There have been several works on streaming algorithms [3, 17, 24, 26]. Others have focused on distributed computing [6, 7, 14, 25]. The work in the distributed setting has focused on algorithms which are implementable in MapReduce, but are also inherently parallel and work in virtually any distributed computing framework. The work of [14] was the first to consider $k$-center clustering in the distributed setting. Their work gave an $O(1)$-round $O(1)$-approximate MapReduce algorithm. Their algorithm is a sampling based MapReduce algorithm which can be used for a variety of clustering objectives. Unfortunately, as the authors point out in their paper, the algorithm does not always perform well empirically for the $k$-center objective since the objective function is very sensitive to missing data points and the sampling can cause large errors in the solution.

The work of Kumar et al. [23] gave a $(1 - \frac{1}{e})$-approximation algorithm for submodular function maximization subject to a cardinality constraint in the MapReduce setting, however, their algorithm requires a non-constant number of MapReduce rounds. Whereas, Mirzasoleiman et al. [25] (recently, extended in [8]) gave a two MapReduce rounds algorithm but their approximation ratio is not constant. It is known that an exact algorithm for submodular maximization subject to a cardinality constraint gives an exact algorithm for the $k$-center problem. Unfortunately, both problems are NP-Hard and the reduction is not approximation preserving. Therefore, their theoretical results do not imply a nontrivial approximation for the $k$-center problem.

For these problems, the following questions loom: What can be achieved for $k$-center clustering with or without outliers in the large-scale distributed setting? What underlying algorithmic ideas are needed for the $k$-center with outliers problem to be solved in the distributed setting? The $k$-center with outliers problem has not been studied in the distributed setting. Given the complexity of the sequential algorithm, it is not clear what such an algorithm would look like.

**Contributions.** In this work, we consider the *k-center* and *k-center with outliers* problems in the distributed computing setting. Although the algorithms are highly parallel and work in virtually any distributed computing framework, they are particularly well suited for the MapReduce [13] as they require only small amounts of inter-machine communication and very little memory on each machine. We therefore state our results for the MapReduce framework [13]. We will assume throughout the paper that our algorithm is given some number of machines, $m$, to process the data. We first begin by considering a natural interpretation of the algorithm of Mirzasoleiman et al. [25] on submodular optimization for the $k$-center problem. The algorithm we introduce runs in two MapReduce rounds and achieves a small constant approximation.

**Theorem 1.1.** *There is a two round MapReduce algorithm which achieves a 4-approximation for the $k$-center problem which communicates $O(km)$ amount of data assuming the data is already partitioned across the machines. The algorithm uses $O(\max\{n/m, mk\})$ memory on each machine.*

Next we consider the *k-center with outliers* problem. This problem is far more challenging and previous distributed techniques do not lend themselves to this problem. Here we combine the algorithm developed for the problem without outliers with the sequential algorithm for $k$-center with outliers. We show a two round MapReduce algorithm that achieves an $O(1)$-approximation.

**Theorem 1.2.** *There is a two round MapReduce algorithm which achieves a 13-approximation for the $k$-center with outliers problem which communicates $O(km \log n)$ amount of data assuming the data is already partitioned across the machines. The algorithm uses $O(\max\{n/m, m(k+z) \log n\})$ memory on each machine.*

Finally, we perform experiments with both algorithms on real world datasets. For *k-center* we observe that the quality of the solutions is effectively the same as that of the sequential algorithm for all values of $k$—the best one could hope for. For the *k-center problem with outliers* our algorithm matches the sequential algorithm as the values of $k$ and $z$ vary and it significantly outperforms the algorithm which does not explicitly consider outliers. Somewhat surprisingly our algorithm achieves an order of magnitude speed-up over the sequential algorithm *even if it is run sequentially*.

## 2   Preliminaries

**Map-Reduce.** We will consider algorithms in the distributed setting where our algorithms are given $m$ machines. We define our algorithms in a general distributed manner, but they particularly suited to the MapReduce model [21]. This model has become widely used both in theory and in applied machine learning [4, 5, 9, 12, 15, 21, 25, 27, 31]. In the MapReduce setting, algorithms run in rounds. In each round the machines are allowed to run a sequential computation without machine communication. Between rounds, data is distributed amongst the machines in preparation for new computation. The goal is to design an algorithm which runs in a small number of rounds since the main running time bottleneck is distributing the data amongst the machine between each round. Generally it is assumed that each of the machines uses sublinear memory [21]. The motivation here is that since MapReduce is used to process large data sets, the memory on the machines should be much smaller than the input size to the problem. It is additionally assumed that there is enough memory to store the entire dataset across all machines. Our algorithms fall into this category and the memory required on each machine scales inversely with $m$.

$k$**-center (with outliers) problem.** In the problems considered, there is a universe $U$ of $n$ points. Between each pair of points $u, v \in U$ there is a distance $d(u, v)$ specifying their dissimilarity. The points are assumed to lie in a metric space which implies that for all $u, v, u' \in U$ we have that 1. $d(u, u) = 0$, 2. $d(u, v) = d(v, u)$, and 3. $d(u, v) \leq d(u, u') + d(u', v)$ *(triangle inequality)*. For a set $X$ of points, we let $d_X(u) := \min_{v \in X}\{d(u, v)\}$ denote the minimum distance of a point $u \in U$ to any point in $X$. In the $k$-center problem, the goal is to choose a set of centers $X$ of $k$ points such that $\max_{v \in U} d_X(v)$ is minimized (i.e., $d_X(v)$ is the distance between $v$ and its cluster center and we would like to minimize the largest distance, across all points). In the $k$-center with outliers problem, the goal is to choose a set $X$ of $k$ points and a set $Z$ of $z$ points such that $\max_{v \in U \setminus Z} d_X(v)$ is minimized. Note that in this problem the algorithm simply needs to choose the set $X$ as the optimal set of $Z$ points is well defined: It is the set of points in $U$ farthest from the centers $X$.

**Sequential algorithms** The most widely used $k$-center (*without* outliers) algorithm is the following simple greedy procedure, summarized in pseudo-code in Algorithm 1. The algorithm sets $X = \emptyset$ and then iteratively adds points from $U$ to $X$ until $|X| = k$. At each step, the algorithm greedily selects the farthest point in $U$ from $X$, and then adds this point to the updated set $X$. This algorithm is natural and efficient and is known to give a 2-approximation for the $k$-center problem [20]. However, it is also inherently sequential and does not lend itself to the distributed setting (except for very small

---
**Algorithm 1** Sequential $k$-center

GREEDY$(U, k)$

1: $X = \emptyset$
2: Add any point $u \in U$ to $X$
3: **while** $|X| < k$ **do**
4:    $u = \mathrm{argmax}_{v \in U}\ d_X(v)$
5:    $X = X \cup \{u\}$
6: **end while**

---

$k$). A naïve MapReduce implementation can be obtained by finding the element $v \in U$ to maximize $d_X(v)$ in a distributed fashion (line 4 in Algorithm 1). This, however, requires $k$ rounds of Map-Reduce that must distribute the entire dataset each round. Therefore it is unsuitably inefficient for many real world problems. The sequential algorithm for $k$-center with outliers is more complicated due to the increased difficulty of the problem (for reference see: [10]). This algorithm is even more fundamentally sequential than Algorithm 1.

## 3   $k$-Center in MapReduce

In this section we consider the $k$-center problem where no outliers are allowed. As mentioned before, a similar variant of this problem has been previously studied in Mirzasoleiman et al. [25] in the distributed setting. The work of Mirzasoleiman et al. considers submodular maximization and showed a $\min\{\frac{1}{k}, \frac{1}{m}\}$-approximation where $m$ is the number of machines. Their algorithm was shown to perform extremely well in practice (in a slightly modified clustering setup). The

$k$-center problem can be mapped to submodular maximization, but the standard reduction is not approximation preserving and their result does not imply a non-trivial approximation for $k$-center. In this section, we give a natural interpretation of their algorithm without submodular maximization.

Algorithm 2 summarizes a distributed approach for solving the k-center problem. First the data points of $U$ are partitioned across all $m$ machines. Then each machine $i$ runs the GREEDY algorithm on the partition they are given to compute a set $C_i$ of $k$ points. These points are assigned to a single machine, which runs GREEDY again to compute the final solution. The algorithm runs in two MapReduce rounds and the only information communicated is $C_i$ for each $i$ if the data is already assigned to machines. Thus, we have the following proposition.

**Proposition 3.1.** *The algorithm* GREEDY-MR *runs in two MapReduce rounds and communicates* $O(km)$ *amount of data assuming the data is originally partitioned across the machines. The algorithm uses* $O(\max\{n/m, mk\})$ *memory on each machine.*

We aim to bound the approximation ratio of GREEDY-MR. Let OPT denote the optimal solution value for the $k$-center problem. The previous proposition and following lemma give Theorem 1.1.

**Lemma 3.2.** *The algorithm* GREEDY-MR *is a* 4-*approximation algorithm.*

**Algorithm 2** Distributed $k$-center

GREEDY-MR$(U, k)$
1: Partition $U$ into $m$ equal sized sets $U_1, \ldots, U_m$ where machine $i$ receives $U_i$.
2: Machine $i$ assigns $C_i = $ GREEDY$(U_i, k)$
3: All sets $C_i$ are assigned to machine 1
4: Machine 1 sets $X = $ GREEDY$(\cup_{i=1}^m C_i, k)$
5: Output $X$

*Proof.* We first show for any $i$ that $d_{C_i}(u) \leq$ 2OPT for any $u \in U_i$. Indeed, say that this is not the case for sake of contradiction for some $i$. Then for some $u \in U_i$, $d_{C_i}(u) > $ 2OPT which implies $u$ is distance greater than 2OPT from all points in $C_i$. By definition of GREEDY for any pair of points $v, v' \in C_i$ it must be the case that $d(v, v') \geq d_{C_i}(u) > $ 2OPT (otherwise $u$ would have been included in $C_i$). Thus, in the set $\{u\} \cup C_i$ there are $k + 1$ points all of distance greater than 2OPT from each other. However, then two of these points $v, v' \in (\{u\} \cup C_i)$ must be assigned to the same center $v^*$ in the optimal solution. Using the triangle inequality and the definition of OPT it must be the case that $d(v, v') \leq d(v^*, v) + d(v^*, v') \leq $ 2OPT, a contradiction. Thus, for all points $u \in U_i$, it must be that $d_{C_i}(u) \leq $ 2OPT.

Let $X$ denote the output solution by GREEDY-MR. We can show a similar result for points in $\cup_{i=1}^m C_i$ when compared to $X$. That is, we show that $d_X(u) \leq $ 2OPT for any $u \in \cup_{i=1}^m C_i$. Indeed, say that this is not the case for sake of contradiction. Then for some $u \in \cup_{i=1}^m C_i$, $d_X(u) > $ 2OPT which implies $u$ is distance greater than 2OPT from all points in $X$. By definition of GREEDY for any pair of points $v, v' \in \cup_{i=1}^m C_i$ it must be that $d(v, v') \geq d_X(u) > $ 2OPT. Thus, in the set $\{u\} \cup X$ there are $k+1$ points all of distance greater than 2OPT from each other. However, then two of these points $v, v' \in (\{u\} \cup X)$ must be assigned to the same center $v^*$ in the optimal solution. Using the triangle inequality and the definition of OPT it must be the case that $d(v, v') \leq d(v^*, v) + d(v^*, v') \leq $ 2OPT, a contradiction. Thus, for all points $u \in \cup_{i=1}^m C_i$, it must be that $d_X(u) \leq $ 2OPT.

Now we put these together to get a 4-approximation. Consider any point $u \in U$. If $u$ is in $C_i$ for any $i$, it must be the case that $d_X(u) \leq $ 2OPT by the above argument. Otherwise, $u$ is not in $C_i$ for any $i$. Let $U_j$ be the partition which $u$ belongs to. We know that $u$ is within distance 2OPT to some point $v \in C_j$ and further we know that $v$ is within distance 2OPT from $X$ from the above arguments. Thus, using the triangle inequality, $d_X(u) \leq d(u, v) + d_X(v) \leq $ 2OPT + 2OPT $\leq $ 4OPT. $\square$

## 4 $k$-center with Outliers

In this section, we consider the $k$-center with outliers problem and give the first MapReduce algorithm for the problem. The problem is more challenging than the version without outliers because one has to also determine which points to discard, which can drastically change which centers should be chosen. Intuitively, the right algorithmic strategy is to choose centers such that there are many points around them. Given that they are surrounded by many points, this is a strong indicator that these points are not outliers. This idea was formalized in the algorithm of Charikar et al. [10], a well-known and influential algorithm for this problem in the single machine setting.

Algorithm 3 summarizes the approach of Charikar et al. [10]. It takes as input the set of points $U$, the desired number centers $k$ and a parameter $G$. The parameter $G$ is a 'guess' of the optimal solution's value. The algorithm's performance is best when $G = $ OPT where OPT denotes the

optimal $k$-center objective after discarding $z$ points. The number of outliers to be discarded, $z$, is not a parameter of the algorithm and is communicated implicitly through $G$. The value of $G$ can be determined by doing a binary search on possible values of $G$—between the minimum and maximum distances of any two points.

For each point $u \in U$, the set $B_u$ contains points within distance $G$ of $u$. The algorithm adds the point $v'$ to the solution set which covers the largest number of points with $B_{v'}$. The idea here is to add points which have many points nearby (and thus large $B_{v'}$). Then the algorithm removes all points from the universe which are within distance $3G$ from $v'$ and continues until $k$ points are chosen to be in the set $X$. Recall that in the outliers problem, choosing the centers is a well

---
**Algorithm 3** Sequential $k$-center with outliers [10]

OUTLIERS$(U, k, G)$

1: $U' = U, X = \emptyset$
2: **while** $|X| < k$ **do**
3:     $\forall u \in U'$ let $B_u = \{v : v \in U', d_{u,v} \leq G\}$
4:     Let $v' = \text{argmax}_{u \in U'} |B_u|$
5:     Set $X = X \cup \{v'\}$
6:     Compute $B'_{v'} = \{v : v \in U', d_{v',v} \leq 3G\}$
7:     $U' = U' \setminus B'_{v'}$
8: **end while**

---

defined solution and the outliers are simply the farthest $z$ points from the centers. Further, it can be shown that when $G = OPT$, after selecting the $k$ centers, there are at most $z$ outliers remaining in $U'$. It is known that this algorithm gives a 3-approximation [10]—however it is not efficient on large or even medium sized datasets due to the computation of the sets $B_u$ within each iteration. For instance, it can take $\approx 100$ hours on a data set with $45,000$ points.

We now give a distributed approach (Algorithm 4) for clustering with outliers. This algorithm is naturally parallel, yet it is significantly faster even if run sequentially on a single machine. It uses a sub-procedure (Algorithm 5) which is a generalization of OUTLIERS.

The algorithm first partitions the points across the $m$ machines (e.g., set $U_i$ goes to machine $i$). Each machine $i$ runs the GREEDY algorithm on $U_i$, but selects $k+z$ points rather than $k$. This results in a set $C_i$. For each $c \in C_i$, we assign a weight $w_c$ that is the number of points in $U_i$ that have $c$ as their closest point in $C_i$ (i.e., if $C_i$ defines an intermediate clustering of $U_i$ then $w_c$ is the number of points in the $c$-cluster). The algorithm then runs a variation of OUTLIERS called CLUSTER, described in Algorithm 5, on only the points in $\cup_{i=1}^m C_i$. The main differences are that

---
**Algorithm 4** Distributed $k$-center with outliers

OUTLIERS-MR$(U, k, z, G, \alpha, \beta)$

1: Partition $U$ into $m$ equal sized sets $U_1, \ldots, U_m$ where machine $i$ receives $U_i$.
2: Machines $i$ sets $C_i = \text{GREEDY}(U_i, k + z)$
3: For each point $c \in C_i$, machine $i$ set $w_c = |\{v : v \in U_i, d(v, c) = d_{C_i}(v)\}| + 1$
4: All sets $C_i$ are assigned to machine 1 with the weights of the points in $C_i$
5: Machine 1 sets $X = \text{CLUSTER}(\cup_{i=1}^m C_i, k, G)$
6: Output $X$

---

CLUSTER represents each point $c$ by the number of points $w_c$ closest to it, and that it uses $5G$ and $11G$ for the radii in $B_u$ and $B'_u$.

The total machine-wise communication required for OUTLIERS-MR is that needed to send each of the sets $C_i$ to Machine 1 along with their weights. Each weight can have size at most $n$, so it only requires $O(\log n)$ space to encode the weight. This gives the following proposition.

**Proposition 4.1.** OUTLIERS-MR *runs in two MapReduce rounds and communicates* $O((k + z)m \log n)$ *amount of data assuming the data is originally partitioned across the machines. The algorithm uses* $O(\max\{n/m, m(k + z) \log n\})$ *memory on each machine.*

---
**Algorithm 5** Clustering subroutine

CLUSTER$(U, k, G)$

1: $U' = U, X = \emptyset$
2: **while** $|X| < k$ **do**
3:     $\forall u \in U$ compute $B_u = \{v : v \in U', d_{u,v} \leq 5G\}$
4:     Let $v' = \text{argmax}_{u \in U} \sum_{u' \in B_u} w_{u'}$
5:     Set $X = X \cup \{v'\}$
6:     Compute $B'_{v'} = \{v : v \in U', d_{v',v} \leq 11G\}$
7:     $U' = U' \setminus B'_{v'}$
8: **end while**
9: Output $X$

---

Our goal is to show that OUTLIERS-MR is an $O(1)$-approximation algorithm (Theorem 1.2). We first present intermediate lemmas and give proof sketches, leaving intermediate proofs to the supplementary material. We overload notation and let OPT denote a fixed optimal solution as well as

the optimal objective to the problem. We will assume throughout the proof that $G = \text{OPT}$, as we can perform a binary search to find $\hat{G} = \text{OPT}(1 + \epsilon)$ for arbitrarily small $\epsilon > 0$ when running CLUSTER on a single machine. We first claim that any point in $U_i$ is not too far from any point in $C_i$.

**Lemma 4.2.** *For every point $u \in U_i$ it is the case that $d_{C_i}(u) \leq 2\text{OPT}$ for all $1 \leq i \leq m$.*

Given the above lemma, let $O_1, \ldots, O_k$ denote the clusters in the optimal solution. A cluster in OPT is defined as a subset of the points in $U$, not including outliers identified by OPT, that are closest to some fixed center chosen by OPT. The high level idea of our proof is similar to that used in [10]. Our goal is to show that when our algorithm choses each center, the set of points discarded from $U'$ in CLUSTER can be mapped to some cluster in the optimal solution. At the end of CLUSTER there should be at most $z$ points in $U'$, which are the outliers in the optimal solution. Knowing that we only discard points from $U'$ close to centers we choose, this will imply the approximation bound.

For every point $u \in U$, which must fall into some $U_i$, we let $c(u)$ denote the closest point in $C_i$ to $u$ (i.e., $c(u)$ is the closest intermediate cluster center found by GREEDY to $u$). Consider the output of CLUSTER, $X = \{x_1, x_2, \ldots, x_k\}$, ordered by how elements were added to $X$. We will say that an optimal cluster $O_i$ is *marked* at CLUSTER iteration $j$ if there is a point $u \in O_i$ such that $c(u) \notin U'$ just before $x_j$ is added to $X$. Essentially if a cluster is marked, we can make no guarantee about covering it within some radius of $x_j$ (which will then be discarded). Figure 1 shows examples where $O_i$ is (and is not) marked. We begin by noting that when $x_j$ is added to $X$ that the weight of the points removed from $U'$ is at least as large as the maximum number of points in an *unmarked* cluster in the optimal solution.

**Lemma 4.3.** *When $x_j$ is added, then $\sum_{u' \in B_{x_j}} w_{u'} \geq |O_i|$ for any unmarked cluster $O_i$.*

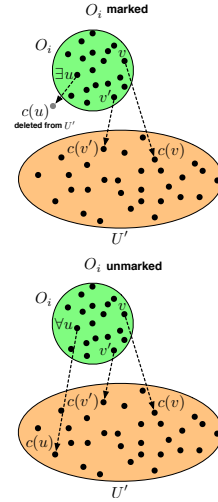

Figure 1: Examples in which $O_i$ is/is not marked.

Given this result, the following lemma considers a point $v$ that is in some cluster $O_i$. If $c(v)$ is within the ball $B_{x_j}$ for $x_j$ added to $X$, then intuitively, this means that we cover all of the points in $O_i$ with $B'_{x_j}$. Another way to say this is that after we remove the ball $B'_{x_j}$, no points in $O_i$ contribute weight to any point in $U'$.

**Lemma 4.4.** *Consider that $x_j$ is to be added to $X$. Say that $c(v) \in B_{x_j}$ for some point $v \in O_i$ for some $i$. Then, for every point $u \in O_i$ either $c(u) \in B'_{x_j}$ or $c(u)$ has already been removed from $U'$.*

See the supplementary material for the proof. The final lemma below states that the weight of the points in $\cup_{x_i : 1 \leq i \leq k} B'_{x_i}$ is at least as large as the number of points in $\cup_{1 \leq i \leq k} O_i$. Further, we know that $|\cup_{1 \leq i \leq k} O_i| = n - z$ since OPT has $z$ outliers. Viewing the points in $B'_{x_i}$ as being assigned to $x_i$ in the algorithm's solution then this shows that the number of points covered is at least as large as the number of points that the optimal solution covers. Hence, there cannot be more than $z$ points uncovered by our algorithm.

**Lemma 4.5.** $\sum_{i=1}^{k} \sum_{u \in B'_{x_i}} w_u \geq n - z$

Finally, we are ready to complete the proof of Theorem 1.2.

**Proof of** [Theorem 1.2] Lemma 4.5 implies that the sum of the weights of the points which are in $\cup_{x_i : 1 \leq i \leq k} B'_{x_i}$ are at least $n - z$. We know that every point $u$ contributes to the weight of some point $c(u)$ which is in $C_i$ for $1 \leq i \leq m$ and by Lemma 4.2 $d(u, c(u)) \leq 2\text{OPT}$. We map every point $u \in U$ to $x_i$, such that $c(u) \in B'_{x_i}$. By definition of $B'_{x_i}$ and Lemma 4.2 it is the case $d(u, x_i) \leq 13\text{OPT}$ by the triangle inequality. Thus, we have mapped $n - z$ points to some point in $X$ within distance 13OPT. Hence, our algorithm discards at most $n - z$ points and achieves a 13-approximation. With Proposition 4.1 we have shown Theorem 1.2. $\square$

## 5   Experiments

We evaluate the real-world performance of the above clustering algorithms on seven clustering datasets, described in Table 1. We compare all methods using the $k$-center with outliers objective, in which $z$ outliers may be discarded. We begin with a brief description of the clustering methods we

Table 1: The clustering datasets (and their descriptions) used for evaluation.

| name | description | $n$ | dim. |
|---|---|---|---|
| *Parkinsons* [28] | patients with early-stage Parkinson's disease | $5,875$ | $22$ |
| *Census*[1] | census household information | $45,222$ | $12$ |
| *Skin*[1] | RGB-pixel samples from face images | $245,057$ | $3$ |
| *Yahoo* [11] | web-search ranking dataset (features are GBRT outputs [29]) | $473,134$ | $500$ |
| *Covertype*[1] | a forest cover dataset with cartographic features | $522,911$ | $13$ |
| *Power*[1] | household electric power readings | $2,049,280$ | $7$ |
| *Higgs*[1] | particle detector measurements (the seven 'high-level' features) | $11,000,000$ | $7$ |

compare. We then show how the distributed algorithms compare with their sequential counterparts on datasets small enough to run the sequential methods, for a variety of settings. Finally, in the large-scale setting, we compare all distributed methods for different settings of $k$.

**Methods.** We implemented the sequential GREEDY and OUTLIERS and distributed GREEDY-MR [25] and OUTLIERS-MR. We also implemented two baseline methods: RANDOM|RANDOM: $m$ machines randomly select $k+z$ points, then a single machine randomly selects $k$ points out of the previously selected $m(k+z)$ points; RANDOM|OUTLIERS: $m$ machines randomly select $k+z$ points, then OUTLIERS (Algorithm 4) is run over the $m(k+z)$ points previously selected; All methods were implemented in MATLAB[TM] and conducted on an 8-core Intel Xeon 2 GHz machine.

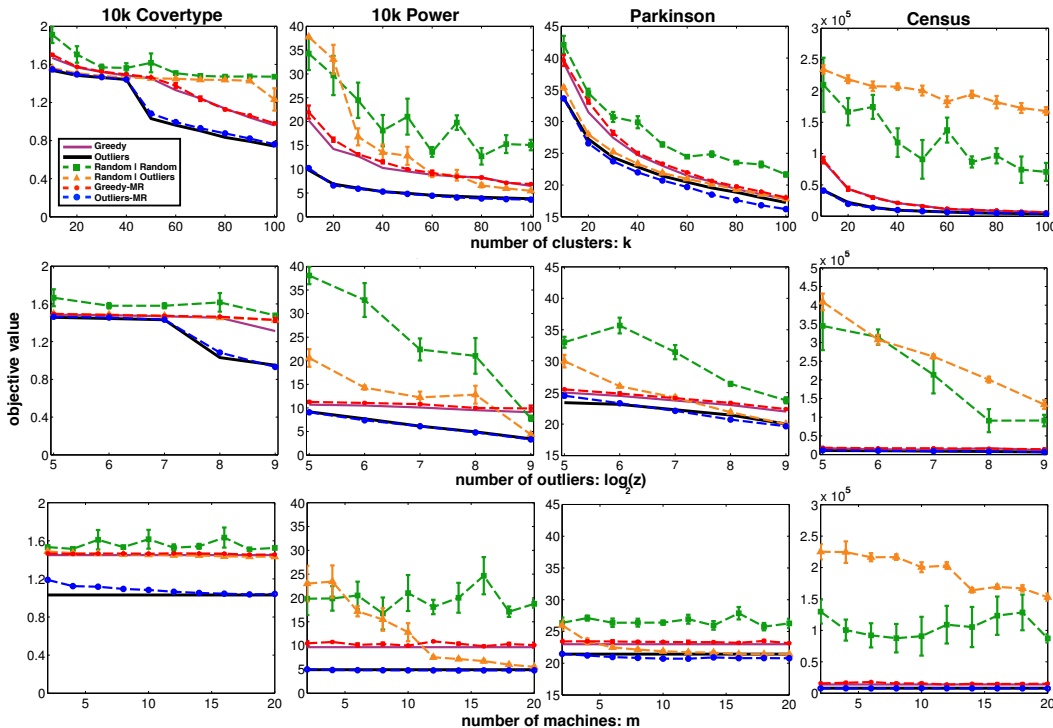

Figure 2: The performance of sequential and distributed methods. We plot the objective value of four small datasets for varying $k$, $z$, and $m$.

**Sequential vs. Distributed.** Our first set of experiments evaluate how close the proposed distributed methods are to their sequential counterparts. To this end, we vary all parameters: number of centers $k$, number of outliers $z$, and the number of machines $m$. We consider datasets for which computing the sequential methods is practical: *Parkinsons*, *Census* and two random subsamples ($10,000$ inputs each) of *Covertype* and *Power*. We show the results in Figure 2. Each column contains the results for a single dataset and each row for a single varying parameter ($k$, $z$, or $m$), along with standard errors over 5 runs. When a parameter is not varied we fix $k=50$, $z=256$, and $m=10$. As expected, the objective value for all methods generally decreases as $k$ increases (as the distance of any point to its cluster center must shrink with more clusters). RANDOM|RANDOM and RANDOM|OUTLIERS usually perform worse than GREEDY-MR for small $k$ (save 10*k Covertype*) and RANDOM|OUTLIERS

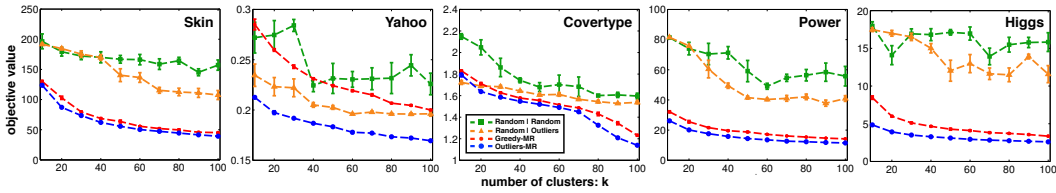

Figure 3: The objective value of five large-scale datasets, for varying $k$

sometimes matches it for large $k$. For all values of $k$ tested, OUTLIERS-MR outperforms all other distributed methods. Furthermore, it matches or slightly outperforms (which we attribute to randomness) the sequential OUTLIERS method in all settings. As $z$ increases the two random methods improve, beyond GREEDY-MR in some cases. Similar to the first plot, OUTLIERS-MR outperforms all other distributed methods while matching the sequential clustering method. For very small settings of $m$ (i.e., $m = 2, 6$), OUTLIERS-MR and GREEDY-MR perform slightly worse than sequential OUTLIERS and GREEDY. However, for practical settings of $m$ (i.e., $m \geq 10$), OUTLIERS-MR matches OUTLIERS and GREEDY-MR matches GREEDY. In terms of speed, on the largest of these datasets (*Census*) OUTLIERS-MR *run sequentially* is more than $677\times$ faster than OUTLIERS, see Table 2. This large speedup is due to the fact that we cannot store the full distance matrix for *Census*, thus all distances need to be computed on demand.

**Large-scale.** Our second set of experiments focus on the performance of the distributed methods on five large-scale datasets, shown in Figure 3. We vary $k$ between $0$ and $100$, and fix $m = 10$ and $z = 256$. Note that for certain datasets, clustering while taking into account outliers produces a noticeable reduction in objective value. On *Yahoo*, the GREEDY-MR method is even outperformed by RANDOM|OUTLIERS that considers outliers. Similar to the small dataset results

Table 2: The speedup of the distributed algorithms, run sequentially, over their sequential counterparts on the small datasets.

| dataset | $k$-center | outliers |
|---|---|---|
| 10*k Covertype* | 3.6 | 6.2 |
| 10*k Power* | 4.8 | 9.4 |
| *Parkinson* | 4.9 | 4.4 |
| *Census* | 12.4 | 677.7 |

OUTLIERS-MR outperforms nearly all distributed methods (save for small $k$ on *Covertype*). Even on datasets where there appear to be few outliers OUTLIERS-MR has excellent performance. Finally, OUTLIERS-MR is extremely fast: clustering on *Higgs* took less than 15 minutes.

## 6 Conclusion

In this work we described algorithms for the $k$-center and $k$-center with outliers problems in the distributed setting. For both problems we studied two round MapReduce algorithms which achieve an $O(1)$-approximation and demonstrated that they perform almost identically to their sequential counterparts on real data. Further, a number of our experiments validate that using $k$-center clustering on noisy data degrades the quality of the solution. We hope these techniques lead to the discovery of fast and efficient distributed algorithms for other clustering problems. In particular, what can be shown for the $k$-median or $k$-means with outliers problems are exciting open questions.

**Acknowledgments** GM was supported by CAPES/BR; MJK and KQW were supported by the NSF grants IIA-1355406, IIS-1149882, EFRI-1137211; and BM was supported by the Google and Yahoo Research Awards.

## Footnotes

[1]https://archive.ics.uci.edu/ml/datasets/

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
