[Supplementary Material]

# Supplementary Material for Fast Distributed $k$-Center Clustering with Outliers on Massive Data

**Gustavo Malkomes,  Matthew J. Kusner,  Wenlin Chen**
Department of Computer Science and Engineering
Washington University in St. Louis
St. Louis, MO 63130
`luizgustavo, mkusner, wenlinchen @wustl.edu`

**Kilian Q. Weinberger**
Department of Computer Science
Cornell University
Ithaca, NY 14850
`kqw4@cornell.edu`

**Benjamin Moseley**
Department of Computer Science and Engineering
Washington University in St. Louis
St. Louis, MO 63130
`bmoseley@wustl.edu`

Here we give the omitted proofs of intermediate results left out of the main text of the paper as well as the complexity analysis of the algorithms.

## $k$-center with Outliers

**Proof of** [Lemma 4.2] For the sake of contradiction say the lemma is not true for some point $u \in U_i$ for some fixed $i$. By definition of GREEDY for any pair of points $v, v' \in C_i$ it must be the case that $d(v, v') \geq d_{C_i}(u) > 2\text{OPT}$. Thus, in the set $\{u\} \cup C_i$ there are $k + z + 1$ points all of distance greater than 2OPT from each other. However, then two of these points $v, v' \in (\{u\} \cup C_i)$ must be assigned to the same center $v^*$ in the optimal solution because OPT can discard at most $z$ of these points. Using the triangle inequality and the definition of OPT it must be the case that $d(v, v') \leq d(v^*, v) + d(v^*, v') \leq 2\text{OPT}$, a contradiction. Thus, for all points $u \in U_i$, it must be that $d_{C_i}(u) \leq 2\text{OPT}$. $\qquad\square$

**Proof of** [Lemma 4.3] Consider any unmarked cluster $O_i$ at time $j$. Since the cluster is unmarked, it is the case that $c(v) \in U'$ just before $x_j$ is added to $X$ for any point $v \in O_i$. Fix a point $u$ where $u := c(v^*)$ for the optimal solution's center $v^*$ that defined $O_i$. Our goal is first to show that $u$ is distance at most 5OPT from $c(v)$ for any point $v \in O_i$.

To bound this, it suffices to bound the sum of the distance of $u$ to any point $v$ in $O_i$ and $v$ to $c(v)$ by the triangle inequality. We know that any point $v \in O_i$ is distance at most OPT from $v^*$ by definition of OPT. Thus, $u$ is distance at most 2OPT from $v^*$ by definition of $u$ and Lemma 4.2 and therefore 3OPT from any point

Figure 1: The 5OPT bound in Lemma 4.3.

in $O_i$ by the triangle inequality. Further, by Lemma 4.2 every point $v \in O_i$ is distance at most 2OPT from $c(v)$. Thus, $u$ is distance at most 5OPT from any point $c(v)$ for $v \in O_i$.

Now we show that $\sum_{v \in B_u} w_v \geq |O_i|$. This is because, every point $c(v)$ must be in $B_u$ by definition of $B_u$ and the fact that $d(u, c(v)) \leq 5\text{OPT}$ by the above argument. Further, every point in $v \in O_i$

contributes to the weight of $c(v)$. Knowing that our algorithm always choses the point $x_j$ such that $\sum_{u \in B_{x_j}} w_u$ is maximized, this completes the proof. $\square$

**Proof of** [Lemma 4.4] Fix some cluster $O_i$ in the optimal solution and some point $v \in O_i$ where $c(v) \in B_{x_j}$. Note that it suffices to prove that $d(x_j, c(u)) \leq 11\text{OPT}$ by definition of $B'_{x_j}$ for any point $u \in O_i$. Fix some point $u$. To prove this, we will use several applications of the triangle inequality. In particular, we will construct a path from $x_j, c(v), v, u, c(u)$. By the triangle inequality, if $d(x_j, c(v)) + d(c(v), v) + d(v, u) + d(u, c(u)) \leq 11\text{OPT}$ then the proof is complete.

Consider $d(x_j, c(v))$ this is at most 5OPT by definition of $B_{x_j}$ and the assumption that $c(v) \in B_{x,j}$. We know that $d(c(v), v) \leq 2\text{OPT}$ and $d(u, c(u)) \leq 2\text{OPT}$ by Lemma 4.2. Finally, we know that $d(u, v) \leq 2\text{OPT}$. This is because both $u$ and $v$ are assigned to the same center in the optimal solution and, therefore, both of them at distance OPT from some point. By triangle inequality, they must be at most 2OPT from each other. Putting this all together completes the proof. $\square$

**Proof of** [Lemma 4.5] To prove the lemma, we will show a one-to-one mapping of each point in $\cup_{1 \leq i \leq k} O_i$ to a unique unit of weight in $\sum_{i=1}^{k} \sum_{u \in B'_{x_i}} w_u$. The proof proceeds by induction. We will show that for any $0 \leq j \leq k$, each unique point in $\cup_{1 \leq i \leq j} O_i$ can be mapped to a unique unit of weight of the points in $\cup_{i=1}^{j} B'_{x_i}$ where $O_1, O_2, \ldots O_j$ is some ordering of the clusters in the optimal solution that we fix inductively.

Assume we have mapped each point in $\cup_{1 \leq i \leq j} O_i$ where $O_1, O_2, \ldots O_j$ for some $0 \leq j \leq k-1$ to a unique unit of weight of the points in $\cup_{i=1}^{j} B'_{x_i}$. Now consider the weight of the points in the set $B'_{x_{j+1}}$. We break the analysis into two cases. For the first case, say that for some $i' \notin [1, 2, \ldots j]$ and $u \in O_{i'}$ it is the case that $c(u)$ is in $B_{x_i}$ for some $i \leq j+1$. Then by Lemma 4.4 it is the case that just after $x_{j+1}$ is added to $X$ all of the points $c(u)$ are no longer in $U'$ for all $u \in O_{i'}$. Thus, in the case, we map each point in $u \in O_{i'}$ to a unit of the weight of $c(u)$. Intuitively, this is the unit of weight that $u$ contributes to $c(u)$.

Figure 2: The 11OPT bound of Lemma 4.4.

Otherwise say that the first case does not hold. Then we know that for all $u \in O_{i'}$ for any $i' \notin [1, 2, \ldots j]$ it is the case that $c(u)$ is in $U'$ after $x_{j+1}$ is added to $X$. By Lemma 4.3 it must be the case that $\sum_{u \in B_{x_{j+1}}} w_u \geq |O_{i'}|$ for any $i' \notin [1, 2, \ldots j]$. In this case, take any cluster in the optimal solution that is not $O_{i'}$ for $1 \leq i \leq j$ and fix this cluster to be $O_{j+1}$. Map each of the points in $O_{j+1}$ to a unique unit of weight of the points in $B'_{x_{j+1}}$. This completely defines the mapping.

Notice that each point must be assigned to a unique unit of weight. This is because, in the first case, each point is charged to the weight it contributes to and then it is removed from $U'$. By removing them from $U'$, we can never later charge to them in the second case. Further, since we charge the points to the weight they contribute to, no two points charge to the same unit of weight in the first case. In the second case, we charge each point $u \in O_{i'}$ to a unit of weight in $B'_{x_{j+1}}$. Knowing that $B'_{x_{j+1}}$ is removed from $U'$, the second case will not charge to this weight again. Further, since $B'_{x_{j+1}}$ does not contain a point $c(u)$ where $u \in O_{i''}$ for any $i'' \notin [1, 2, \ldots j]$, none of the points' weight in $B'_{x_{j+1}}$ comes from a point in $O_{i''}$ for any $i'' \notin [1, 2, \ldots j]$, so no weight of the points in $B'_{x_{j+1}}$ can be charged to by the first case later. $\square$

## Complexity

To assess the complexity of the sequential and distributed methods let us assume that computing a distance between any two inputs requires $O(f)$ time, where $f$ is some distance function complexity.

The complexity of GREEDY, the sequential $k$-center method is then $O(nkf)$ as we have to at most compute the distance between all $n$ points and the $k$ centers. The complexity of the distributed version GREEDY-MR is $O((\frac{n}{m}+mk)kf)$ as there are two rounds of greedy and different sets of points to consider. For $k$-center clustering with outliers, we give the complexity of the case where, before the loops of Algorithms 4 and 5, we can compute the full distance matrix between all inputs, which is $O(n^2f)$. And that for each point $u \in U$, we can sort the remaining points by closeness to $u$, giving a complexity of $O(n^2 \log n)$ For both algorithms, all operations within the loop have smaller order than these initial operations. Therefore, for OUTLIERS the complexity is $O(n^2(\log n + f))$. Finally, for OUTLIERS-MR as we first run GREEDY and then a variation of OUTLIERS the final complexity is $O(\frac{n}{m}(k+z)f + ((k+z)m)^2[\log((k+z)m) + f])$.