[Reviews · NeurIPS 2015]

Submitted by Assigned_Reviewer_1

The 4-approximation is surprisingly simple: simply compute the k-center (approximately) in each machine, collect all such centers in a single machine, and then compute the k-center for these centers. The proof is also very simple. I am not sure if this simple argument has been found (or implied) from previous work. But if not, it is definitely a simple, yet nice and cute new result.

The algorithm for the outlier version is somewhat harder. Again, compute the k-center (approximately) for the points in each machine, and collect all such centers in a single machine. Now, each such center can be through as a representative of several original points (by assigning a proper weight). Then the machine with all such centers runs an algorithm similar to Charikar et al. The analysis of the approximation ratio requires some effort.

Overall, the results of the paper are nice. The theoretical depth is ok, yet not terribly deep nor exciting. I would rate it as a weak accept (7).

Some minor comments:

The idea of creating weighted point is somewhat similar to the recent work for a generalization of the outlier k-center problem. Chen, Danny Z., et al. "Matroid and knapsack center problems."

Integer Programming and Combinatorial Optimization. Springer Berlin Heidelberg, 2013. 110-122.

The work is in the very same spirit of several recent work on distributed and composable coreset, in which the idea is to compute a coreset on each machine, and then compute the approximate optimal solution for the union of all coresets. These papers are definitely related.

Indyk, Piotr, et al. "Composable core-sets for diversity and coverage maximization."

Proceedings of the 33rd ACM SIGMOD-SIGACT-SIGART symposium on Principles of database systems. ACM, 2014. Barbosa, Rafael, et al. "The Power of Randomization: Distributed Submodular Maximization on Massive Datasets."

arXiv preprint arXiv:1502.02606 (2015).
Summary: The paper considers the classic k-center clustering problem in the mapreduce setting. Since k-center is a fundamental clustering algorithm, studying the algorithms in the distributed setting is meaningful.

The main contributions of the paper include a two round mapreduce 4-approximation for k-center and a two rounds O(1) approximation with outliers.

Submitted by Assigned_Reviewer_2

The algorithms and theoretical analysis are interesting and solid. Especially, I like the idea that adapting [9]'s algorithm to distributed k-center with outliers. Communication complexity is also considered in this paper, which is important in distributed computing.

1. The authors said the work is inspired by [23]. I am wondering how much content in this paper is new compared to [23].

2. Is there any lower bound for the communication complexity, especially for the case with outliers.

3. The time complexity is quadratic in the number of outliers z. But in some practical case, like the experiments, the algorithm can only handle small number of outliers, while usually z takes a constant fraction of n, for example 5%, which could be a large number.

4. I think there is a typo in line 178, it should be ``from all points in C_i".
Summary: The paper focused on k-center clustering problem with outliers in a distributed setting, and gave two algorithms for the cases without and with outliers. The algorithms can be efficiently implemented in Map-reduce resulting in constant approximations.

Submitted by Assigned_Reviewer_3

This paper presents two distributed k-center clustering algorithms to work with and without outliers data sets. Theoretical results and extensive experiments are provided.

Map Reduce: The capability to run in parallel is interesting in itself. I could not understand why the authors try to link the algorithms to MapReduce framework. In fact, your summary of MapReduce framework in section 2 is fairly superficial.

Algorithm 4: how do you compute z? I guess z is not very important factor. One can just set it as large as a machine can handle data with its available memory. At step 1, you partition U into m equal subsets. Could you provide insights into how performance could be affected if the subsets do not have same size?

Algorithm 5: It would be good if you could explain intuitively why you use 5G and 11G thresholds.

Theorem 1.2: 13-approximation seems to be very large compared to 2-approximation. Could you explain why such algorithm with 13-approximation is still good to use in practice?

Line 107: why is it a surprise? Should not you be able to predict that?

About the proofs: would not it be tidy to put lemma 4.2 before 3.2 then you do not have to prove this lemma again and again in lemma 3.2?

Experiments: I am wondering why you do not arrange the charts in the order of the data sets listed in table 1.

Summary: This paper discusses two fast distributed k-center clustering algorithms in the settings with and without outliers. Theoretical and experimental results are pretty interesting, though not much interpretation to the impacts of theoretical results is provided.

Author Feedback
Author rebuttal: We thank all reviewers for their insightful comments. We plan to address the suggestions as far as possible in the final version.

-Related work on coresets (Reviewer 1)
Our algorithmic techniques are similar to coresets as the reviewer has pointed out and coresets have been shown to be useful in a large variety of clustering problems. We will cite work on coresets and compare them to our algorithms in a final version of the paper.

-Comparing to the work of [23] (Reviewer 1&2)
Though our algorithms are inspired by [23], we consider substantially different problems and use different analysis techniques. Some other differences are that we show a constant approximation bounds for the k-center problem, which is much tighter than the non-constant bound given in [23]. Second, we address a different and much harder variant of k-center problem where noise is present. To our knowledge, no previous work has proposed a distributed algorithm for the k-center problem with outliers.

-Lower bound on commutation complexity (Reviewer 2)
There are currently no known lower bounds on the communication required for the problems considered. In fact, there are only a couple restrictive lower bounds known for the MapReduce framework studied and it is a central open problem to discover unrestricted lower bounds for any problem in this model.

-The parameter z (Reviewer 2&3)
z is a hyper-parameter in the problem and its application-specific.

-Mapreduce Theoretical framework (Reviewer 3)
Our main focus is to address large-scale k-center problems where one machine cannot handle all the data. Currently, the most popular and well studied theoretical model for distributed computing is the MapReduce model we consider in the paper. While this model is designed for MapReduce, it captures the fundamental constraints seen in distributed computing in general and the model is not restrictive to MapReduce.

-The Approximation ratio (Reviewer 3)
The approximation ratio of 13 is large compared to 2, however the 13 is necessary for technical reasons. We observe that our algorithm does very well experimentally on real datasets and we believe that real data is much better behaved than the worst case instances that could arise in the theoretical analysis. We believe that the proof that the algorithm is a 13-approximation gives insights into why the algorithm works well and additionally shows that the algorithm has bounded performance even in the worst case.

-About presentation (Reviewer 3)
We'll re-organize the text and give more explanation as you suggested. Specifically, about intuition on why 5G and 11G are used in the algorithm, we'll add this to the text. Currently intuition is provided in Figures 1 and 2 of the supplementary material.

-About dimensionality (Reviewer 6)
Our algorithm could be applied on all data with a well-defined distance metric, including high-dimensional data.